# Efficient Certification of Physics-Informed Neural Networks

## Abstract

Recent work provides promising evidence that Physics-Informed Neural Networks (PINN) can efficiently solve partial differential equations (PDE). However, previous works have failed to provide guarantees on the *worst-case* residual error of a PINN across the spatio-temporal domain – a measure akin to the tolerance of numerical solvers – focusing instead on point-wise comparisons between their solution and the ones obtained by a solver on a set of inputs. In real-world applications, one cannot consider tests on a finite set of points to be sufficient grounds for deployment, as the performance could be substantially worse on a different set. To alleviate this issue, we establish tolerance-based *correctness* conditions for PINNs over the *entire* input domain. To verify the extent to which they hold, we introduce $\partial$-CROWN: a general, efficient and scalable post-training framework to bound PINN residual errors. We demonstrate its effectiveness in obtaining tight certificates by applying it to two classically studied PDEs – Burgers' and Schrödinger's equations –, and two more challenging ones with real-world applications – the Allan-Cahn and Diffusion-Sorption equations.

## 1 Introduction

Accurately predicting the evolution of complex systems through simulation is a difficult, yet necessary, process in the physical sciences. Many of these systems are represented by partial differential equations (PDE) the solutions of which, while well understood, pose a major computational challenge to solve at an appropriate spatio-temporal resolution (Raissi et al., 2019a; Kochkov et al., 2020). Inspired by the success of machine learning in other domains, recent work has attempted to overcome the aforementioned challenge through *physics-informed neural networks* (PINN) (Raissi et al., 2019a; Sun et al., 2020; Pang et al., 2019). For example, the Diffusion-Sorption equation – which has real-world applications in the modeling of groundwater contaminant transport – takes 59.83s to solve per inference point using a classical PDE solver, while inference in its PINN version from Takamoto et al. (2022) takes only $2.7 \times 10^{-3}$s, a speed-up of more than $10^4$ times.

The parameters of a PINN are estimated by minimizing the residual of the given PDE, together with its initial and boundary conditions, over a set of spatio-temporal training inputs. Its accuracy is then empirically estimated by measuring the output over separate held-out inputs, and (typically) comparing them to standard numerical PDE solvers. In other words, most current work on PINNs provides no formal correctness guarantees that are applicable for *every* input within the feasible domain. We argue that, while testing on a finite set of points provides a good initial signal on the accuracy of the PINN, such an approach cannot be relied upon in practice, and error certification is needed to understand the quality of the PINN trained (Hillebrecht and Unger, 2022).

In order to alleviate the deficiencies of previous evaluation criteria, we introduce formal, tolerance-based *correctness* conditions for PINNs. These require that the residual error is *globally* upper bounded across the domain by a tolerance parameter. To compute this bound and verify the correctness conditions, we build on the progress that has been made in the neural network verification literature. Specifically, we extend the CROWN framework (Zhang et al., 2018) by deriving linear upper and lower bounds for the various nonlinear terms that appear in PINNs, and devise a novel customized bound propagation strategy for the task at hand.

Our contributions are threefold. **(i)** We formally define global correctness conditions for general PINNs that approximate solutions of PDEs. **(ii)** We introduce a general, efficient, and scalable post-training *correctness certification framework* ($\partial$-CROWN) to theoretically verify PINNs over their entire spatio-temporal domains. **(iii)** We demonstrate our post-training framework on two widely studied PDEs in the context of PINNs, Burgers' and Schrödinger's equations (Raissi et al., 2019a), and two more challenging ones with real-world applications, the Allan-Cahn equation (Monaco and Apiletti, 2023) and the Diffusion-Sorption equation (Takamoto et al., 2022).

## 2  RELATED WORK

Since our certification framework for PINNs is based on the verification literature of image classifiers, in this section we explore: related work for PINNs, and previous work on NN robustness verification.

**Physics-informed Neural Networks**  Dissanayake and Phan-Thien (1994) first discussed using neural networks to approximate PDE solutions under a supervised learning paradigm. More recently, Raissi et al. (2019a) introduced PINNs, which leverage automatic differentiation to obtain approximate solutions to the underlying PDE. Since then, a variety of different PINNs have emerged in a range of applications, from fluid dynamics (Raissi et al., 2019b; 2020; Sun et al., 2020; Jin et al., 2021), to meta material design (Liu and Wang, 2019; Fang and Zhan, 2019a; Chen et al., 2020) for different classes of PDEs (Pang et al., 2019; Fang and Zhan, 2019b; Zhang et al., 2020). A few works analyze the convergence of the training process of PINNs under specific conditions (Shin et al., 2020; Wang et al., 2022b). Mishra and Molinaro (2022) approximated the generalization error of various PINNs under specific stability and training process assumptions, and others introduced approximation bounds under regularity assumptions (Ryck and Mishra, 2022; Hillebrecht and Unger, 2022). Our verification framework is applicable to any PINN where the solution is modeled by a fully connected network.

**Robustness Verification of Neural Networks**  The presence of adversarial examples, *i.e.*, small local input perturbations that lead to large output changes, was established by Szegedy et al. (2013) in image classifiers. As robust classifiers emerged (Madry et al., 2017), so did attempts to certify them formally. Those methods can be divided into *exact*, *i.e.*, complete (Katz et al., 2017; Ehlers, 2017; Huang et al., 2017; Lomuscio and Maganti, 2017; Bunel et al., 2018), or *conservative*, *i.e.*, sound but incomplete (Gowal et al., 2018; Mirman et al., 2018; Wang et al., 2018; Wong and Kolter, 2018; Ayers et al., 2020). A promising set of conservative methods poses the problem as a convex relaxation of the original nonlinear network architecture, and solves it using a linear programming solver (Salman et al., 2019) or by obtaining closed-form bounds (Zhang et al., 2018; Wang et al., 2021). The latter are especially appealing due to their efficiency. Examples include CROWN (Zhang et al., 2018) and $\alpha$-CROWN (Xu et al., 2020b). Xu et al. (2020a) extended the linear relaxation framework from Zhang et al. (2018) to general computation graphs, but the purely backward propagation nature makes it potentially less efficient than custom bounds/hybrid approaches (Shi et al., 2020).

We use techniques from robustness verification typically applied in a local input neighborhoods to certify the *full* applicability domains of PINNs. To the best of our knowledge, ours is the first application of these methods to a *'global'* specification, and within a scientific context.

## 3  PRELIMINARIES

### 3.1  NOTATION

Given vector $\mathbf{a} \in \mathbb{R}^d$, $\mathbf{a}_i$ refers to its $i$-th component. We use $\partial_{\mathbf{x}_i^j} f$ and $\frac{\partial^j f}{(\partial \mathbf{x}_i)^j}$ interchangeably to refer to the $j$-th partial derivative of a function $f : \mathbb{R}^n \to \mathbb{R}$ with respect to the $i$-component of its input, $\mathbf{x}_i$. Where it is clear, we use $f(\mathbf{x})$ and $f$ interchangeably. We take $\mathbb{L}_{\mathbf{W},\mathbf{b}}^{(i)}(\mathbf{x}) = \mathbf{W}^{(i)}\mathbf{x} + \mathbf{b}^{(i)}$ to be a function of $\mathbf{x}$ parameterized by weights $\mathbf{W}^{(i)}$ and bias $\mathbf{b}^{(i)}$. We define an $L$-layer *fully connected neural network* $g : \mathbb{R}^{d_0} \to \mathbb{R}^{d_L}$ for an input $\mathbf{x}$ as $g(\mathbf{x}) = y^{(L)}(\mathbf{x})$ where $y^{(k)}(\mathbf{x}) = \mathbb{L}_{\mathbf{W},\mathbf{b}}^{(k)}(z^{(k-1)}(\mathbf{x}))$, $z^{(k-1)}(\mathbf{x}) = \sigma(y^{(k-1)}(\mathbf{x}))$, $z^{(0)}(\mathbf{x}) = \mathbf{x}$, in which $\mathbf{W}^{(k)} \in \mathbb{R}^{d_k \times d_{k-1}}$ and $\mathbf{b}^{(k)} \in \mathbb{R}^{d_k}$ are the weight and bias of layer $k$, $\sigma$ is the nonlinear activation, and $k \in \{1, \ldots, L\}$.

## 3.2 PHYSICS-INFORMED NEURAL NETWORKS (PINNS)

We consider general nonlinear PDEs of the form:

$$f(t, \hat{\mathbf{x}}) = \partial_t u(t, \hat{\mathbf{x}}) + \mathcal{N}[u](t, \hat{\mathbf{x}}) = 0, \ \hat{\mathbf{x}} \in \mathcal{D}, t \in [0, T], \tag{1}$$

where $f$ is the residual of the PDE, $t$ is the temporal and $\hat{\mathbf{x}}$ is the spatial components of the input, $u : [0, T] \times \mathcal{D} \to \mathbb{R}$ is the solution, $\mathcal{N}$ is a nonlinear differential operator on $u$, $T \in \mathbb{R}^+$, and $\mathcal{D} \subset \mathbb{R}^D$. Where possible, for conciseness we will use $\mathbf{x} = (t, \hat{\mathbf{x}})$, for $\mathbf{x} \in \mathcal{C} = [0, T] \times \mathcal{D}$, with $\mathbf{x}_0 = t$.

We assume $f$ is the residual of an $R^{th}$ order PDE where the differential operators of $\mathcal{N}$ applied to $u$ yield the partial derivatives for order $\{0, ..., R\}$ as: $u \in \mathcal{N}^{(0)}$, $\partial_{\mathbf{x}_i} u \in \mathcal{N}^{(1)}$, $\partial_{\mathbf{x}_i^2} u \in \mathcal{N}^{(2)}$, ..., $\partial_{\mathbf{x}_i^R} u \in \mathcal{N}^{(R)}$ for $i \in \{0, \ldots, D\}$[1]. With these, we can re-write $f = \mathcal{P}(u, \partial_{\mathbf{x}_0} u, \ldots, \partial_{\mathbf{x}_D} u, \ldots, \partial_{\mathbf{x}_D^R} u)$, where $\mathcal{P}$ is a nonlinear function of those terms. Furthermore, the PDE is defined under (1) initial conditions, *i.e.*, $u(0, \hat{\mathbf{x}}) = u_0(\hat{\mathbf{x}})$, for $\hat{\mathbf{x}} \in \mathcal{D}$, and (2) general Robin boundary conditions, *i.e.*, $au(t, \hat{\mathbf{x}}) + b\partial_{\mathbf{n}} u(t, \hat{\mathbf{x}}) = u_b(t, \hat{\mathbf{x}})$ for $a, b \in \mathbb{R}$, $\hat{\mathbf{x}} \in \delta\mathcal{D}$ and $t \in [0, T]$, and $\partial_{\mathbf{n}} u$ is the normal derivative at the border with respect to some components of $\hat{\mathbf{x}}$.

Continuous-time PINNs (Raissi et al., 2019a) result from approximating the solution, $u(\mathbf{x})$, using a neural network parameterized by $\theta$, $u_\theta(\mathbf{x})$. We refer to this network as the *approximate solution*. In that context, the *physics-informed neural network* (or residual) is $f_\theta(\mathbf{x}) = \partial_t u_\theta(\mathbf{x}) + \mathcal{N}[u_\theta](\mathbf{x})$. For example, the one-dimensional Burgers' equation (explored in detail in Section 6) is defined as:

$$f_\theta(\mathbf{x}) = \partial_t u_\theta(\mathbf{x}) + u_\theta(\mathbf{x})\partial_x u_\theta(\mathbf{x}) - (0.01/\pi)\partial_{x^2} u_\theta(\mathbf{x}). \tag{2}$$

Note $f_\theta$ has the same order as $f$, and can be described similarly as a nonlinear function with the partial derivatives applied to $u_\theta$ instead of $u$. For example, Burgers' equation from above has one $0^{th}$ order term ($u_\theta$), two $1^{st}$ order ones ($\partial_t u_\theta$ and $\partial_x u_\theta$), and a $2^{nd}$ order partial derivative ($\partial_{x^2} u_\theta$), while $u_\theta(\mathbf{x})\partial_x u_\theta(\mathbf{x})$ is a nonlinear term of the $f_\theta$ polynomial.

## 3.3 BOUNDING NEURAL NETWORK OUTPUTS USING CROWN (ZHANG ET AL., 2018)

The computation of upper/lower bounds on the output of neural networks over a domain has been widely studied within verification of image classifiers (Katz et al., 2017; Mirman et al., 2018; Zhang et al., 2018). For the sake of computational efficiency, we consider the bounds obtained using CROWN (Zhang et al., 2018)/$\alpha$-CROWN (Xu et al., 2020b) as the base for our framework.

Take $g$ to be the fully connected neural network (as defined in Section 3.1) we're interested in bounding. The goal is to compute $\max / \min_{\mathbf{x} \in \mathcal{C}} g(\mathbf{x})$, where $\mathcal{C}$ is the applicability domain. Typically within verification of image classifiers, $\mathcal{C} = \mathbb{B}_{\mathbf{x}, \epsilon}^p = \{\mathbf{x}' : \|\mathbf{x}' - \mathbf{x}\|_p \le \epsilon\}$, *i.e.*, it is a *local* $\ell_p$-ball of radius $\epsilon$ around an input from the test set $\mathbf{x}$.

CROWN solves the optimization problem by *back-propagating* linear bounds of $g(\mathbf{x})$ through each hidden layer of the network until the input is reached. To do so, assuming constant bounds on $y^{(k)}(\mathbf{x})$ are known for $\mathbf{x} \in \mathcal{C}$, *i.e.*, $\forall \mathbf{x} \in \mathcal{C} : y^{(k),L} \le y^{(k)}(\mathbf{x}) \le y^{(k),U}$, CROWN relaxes the nonlinearities of each $z^{(k)}$ using a linear lower and upper bound approximation that contains the full possible range of $\sigma(y^{(k)}(\mathbf{x}))$. By relaxing the activations of each layer and back-propagating it through $z^{(k)}$, CROWN obtains a bound on each $y^{(k)}$ as a function of $y^{(k-1)}$. Back-substituting from the output $y^{(L)} = g(\mathbf{x})$ until the input $\mathbf{x}$ results in:

$$\min_{\mathbf{x} \in \mathcal{C}} g(\mathbf{x}) \ge \min_{\mathbf{x} \in \mathcal{C}} \mathbf{A}^L \mathbf{x} + \mathbf{a}^L, \ \max_{\mathbf{x} \in \mathcal{C}} g(\mathbf{x}) \le \max_{\mathbf{x} \in \mathcal{C}} \mathbf{A}^U \mathbf{x} + \mathbf{a}^U,$$

where $\mathbf{A}^L$, $\mathbf{a}^L$, $\mathbf{A}^U$ and $\mathbf{a}^U$ are computed in polynomial time from $\mathbf{W}^{(k)}, \mathbf{b}^{(k)}$, and the linear relaxation parameters. The solution to the optimization problems above given simple constraints $\mathcal{C}$ can be obtained in closed-form. $\alpha$-CROWN (Xu et al., 2020b) improves these bounds by optimizing the linear relaxations for tightness.

---

[1]For simplicity, we assume $\mathcal{N}$ does not contain any cross-derivative operators, yet an extension would be trivial to derive.

## 4 CORRECTNESS CONDITIONS FOR PINNs

By definition, $u_\theta$ is a correct solution to the PINN $f_\theta$ – and therefore the PDE $f(\mathbf{x}) = 0$ – if 3 conditions are met: ① the norm of the solution error with respect to the initial condition is upper bounded within an acceptable tolerance, ② the norm of the solution error with respect to the boundary conditions is bounded within an acceptable tolerance, and ③ the norm of the residual is bounded within an acceptable convergence tolerance. We define these as PINN *correctness conditions*, and formalize it in Definition 1.

**Definition 1** (Correctness Conditions for PINNs). *$u_\theta : [0, T] \times \mathcal{D} \to \mathbb{R}$ is a $\delta_0, \delta_b, \varepsilon$-globally correct approximation of the exact solution $u : [0, T] \times \mathcal{D} \to \mathbb{R}$ if:*

$$① \quad \max_{\hat{\mathbf{x}} \in \mathcal{D}} |u_\theta(0, \hat{\mathbf{x}}) - u_0(\hat{\mathbf{x}})|^2 \leq \delta_0,$$

$$② \quad \max_{t \in [0,T], \hat{\mathbf{x}} \in \delta\mathcal{D}} |au_\theta(t, \hat{\mathbf{x}}) + b\partial_{\mathbf{n}} u_\theta(t, \hat{\mathbf{x}}) - u_b(t, \hat{\mathbf{x}})|^2 \leq \delta_b,$$

$$③ \quad \max_{\mathbf{x} \in \mathcal{C}} |f_\theta(\mathbf{x})|^2 \leq \varepsilon.$$

Previous works deriving from Raissi et al. (2019a) have measured the correctness of the approximation $u_\theta$ empirically through the error between $u_\theta$ and a solution obtained via either analytical or numerical solvers for $f$, satisfying a relaxed, empirical version of these conditions only. In practice, $\delta_0$, $\delta_b$, and $\varepsilon$ correspond to tolerances similar to the ones given by numerical solvers for $f$.

## 5 $\partial$-CROWN: PINN CORRECTNESS CERTIFICATION FRAMEWORK

The verification of the PINN correctness conditions from Definition 1 requires bounding a linear function of $u_\theta$ for ①. Moreover, it requires bounds for a linear function of $u_\theta$ and $\partial_{\mathbf{n}} u_\theta$ for ②, and the PINN output, $f_\theta$, in ③. To achieve ①, assuming $u_\theta$ is a standard fully connected neural network as in Raissi et al. (2019a), we can directly use CROWN/$\alpha$-CROWN (Zhang et al., 2018; Xu et al., 2020b). However, bounding ② and ③ with a linear function in $\mathbf{x}$ efficiently requires a method to bound linear and nonlinear functions of the partial derivatives of $u_\theta$.

We propose $\partial$-CROWN, an efficient framework to: (i) compute closed-form bounds on the partial derivatives of an arbitrary fully-connected network $u_\theta$ (Section 5.1), and (ii) bound a nonlinear function of those partial derivative terms, *i.e.*, $f_\theta$ (Section 5.2). Throughout this section, we assume $u_\theta(\mathbf{x}) = g(\mathbf{x})$ as defined in Section 3.1, with $d_0 = 1 + D$. Proofs for lemmas and theorems presented in this section are in Appendix E.

### 5.1 BOUNDING PARTIAL DERIVATIVES OF $u_\theta$

The computation of the bounds for the $0^{th}$ order derivative, *i.e.*, $u_\theta$, and intermediate pre-activations can be done using CROWN/$\alpha$-CROWN (Zhang et al., 2018; Xu et al., 2020b). As such, for what follows, we assume that for $\mathbf{x} \in \mathcal{C}$, both the bounds on $u_\theta$ and $y^{(k)}$, $\forall k$ are given.

**Assumption 1.** *The pre-activation layer outputs of $u_\theta$, $y^{(k)} = \mathbb{L}_{\mathbf{W},\mathbf{b}}^{(k)}(z^{(k-1)})$, are lower and upper bounded by linear functions $\mathbb{L}_{\mathbf{A},\mathbf{a}}^{(k),L}(\mathbf{x}) \leq y^{(k)} \leq \mathbb{L}_{\mathbf{A},\mathbf{a}}^{(k),U}(\mathbf{x})$. Moreover, for $\mathbf{x} \in \mathcal{C}$, we have $y^{(k),L} \leq y^{(k)} \leq y^{(k),U}$.*

Note that using CROWN/$\alpha$-CROWN, $\mathbf{A}^{(k),L}$, $\mathbf{a}^{(k),L}$, $\mathbf{A}^{(k),U}$, $\mathbf{a}^{(k),U}$ are functions of all the previous layers' parameters. For $1^{st}$ order derivatives, we start by explicitly obtaining the expression of $\partial_{\mathbf{x}_i} u_\theta$.

**Lemma 1** (Computing $\partial_{\mathbf{x}_i} u_\theta$). *For $i \in \{1, \ldots, d_0\}$, the partial derivative of $u_\theta$ with respect to $\mathbf{x}_i$ can be computed recursively as $\partial_{\mathbf{x}_i} u_\theta = \mathbf{W}^{(L)} \partial_{\mathbf{x}_i} z^{(L-1)}$ for:*

$$\partial_{\mathbf{x}_i} z^{(k)} = \partial_{z^{(k-1)}} z^{(k)} \partial_{\mathbf{x}_i} z^{(k-1)}, \quad \partial_{\mathbf{x}_i} z^{(0)} = \mathbf{e}_i,$$

*for $k \in \{1, \ldots, L-1\}$, and where $\partial_{z^{(k-1)}} z^{(k)} = diag\left[\sigma'\left(y^{(k)}\right)\right] \mathbf{W}^{(k)}$.*

Using this result, we can efficiently linearly lower and upper bound $\partial_{\mathbf{x}_i} u_\theta$.

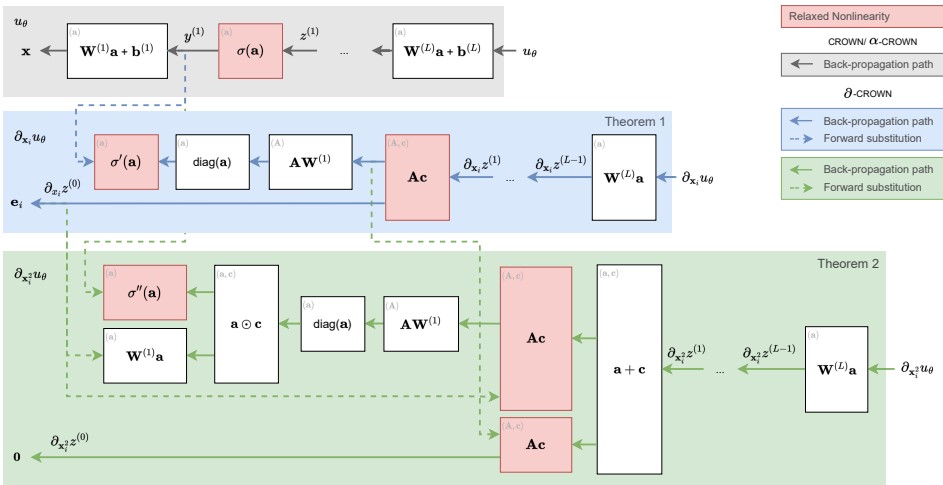

Figure 1: **Bounding Partial Derivatives with $\partial$-CROWN**: our hybrid scheme for bounding $\partial_{\mathbf{x}_i} u_\theta$ and $\partial_{\mathbf{x}_i^2} u_\theta$ uses back-propagation and forward substitution (inspired by Shi et al. (2020)) to compute bounds in $\mathcal{O}(L)$ instead of the $\mathcal{O}(L^2)$ complexity of full back-propagation as in Xu et al. (2020a).

**Theorem 1** ($\partial$-CROWN: Linear Bounding $\partial_{\mathbf{x}_i} u_\theta$). *There exist two linear functions $\partial_{\mathbf{x}_i} u_\theta^U$ and $\partial_{\mathbf{x}_i} u_\theta^L$ such that, $\forall \mathbf{x} \in \mathcal{C}$ it holds that $\partial_{\mathbf{x}_i} u_\theta^L \leq \partial_{\mathbf{x}_i} u_\theta \leq \partial_{\mathbf{x}_i} u_\theta^U$, where the linear coefficients can be computed recursively in closed-form in $\mathcal{O}(L)$ time.*

The formal statement of Theorem 1 and expressions for $\partial_{\mathbf{x}_i} u_\theta^L$ and $\partial_{\mathbf{x}_i} u_\theta^U$ are provided in Appendix E.3. Note that this bound is not computed using fully backward propagation as in Xu et al. (2020a). Instead we use a *hybrid* scheme in the spirit of Shi et al. (2020) for the sake of efficiency. We perform backward propagation to compute $\partial_{z^{(k-1)}} z^{(k)}$ as a function of $y^{(k)}$, and forward-substitute the pre-computed CROWN bounds $\mathbb{L}_{\mathbf{A},\mathbf{a}}^{(k),L}(\mathbf{x}) \leq y^{(k)} \leq \mathbb{L}_{\mathbf{A},\mathbf{a}}^{(k),U}(\mathbf{x})$ at that point instead of fully backward propagating which would have $\mathcal{O}(L^2)$ complexity. This induces a significant speed-up while achieving tight enough bounds. Figure 1 showcases the back-propagation and forward substitution paths for bounding $\partial_{\mathbf{x}_i} u_\theta$ in blue. Similarly to CROWN with the activation $\sigma$, this bound requires relaxing $\sigma'(y^{(k)})$.

Similarly, we can linearly bound $\partial_{\mathbf{x}_i^2} u_\theta$, a requirement to bound $f_\theta$ in $2^{nd}$ order PINNs.

**Lemma 2** (Expression for $\partial_{\mathbf{x}_i^2} u_\theta(\mathbf{x})$). *For $i \in \{1, \ldots, d_0\}$, the second partial derivative of $u_\theta$ with respect to $\mathbf{x}_i$ can be computed recursively as $\partial_{\mathbf{x}_i^2} u_\theta = \mathbf{W}^{(L)} \partial_{\mathbf{x}_i^2} z^{(L-1)}$ where:*

$$\partial_{\mathbf{x}_i^2} z^{(k)} = \partial_{x_i z^{(k-1)}} z^{(k)} \partial_{\mathbf{x}_i} z^{(k-1)} + \partial_{z^{(k-1)}} z^{(k)} \partial_{\mathbf{x}_i^2} z^{(k-1)},$$

*and $\partial_{\mathbf{x}_i^2} z^{(0)} = \mathbf{0}$, for $k \in \{1, \ldots, L-1\}$, with $\partial_{\mathbf{x}_i} z^{(k-1)}$ and $\partial_{z^{(k-1)}} z^{(k)}$ as per in Lemma 1, and $\partial_{x_i z^{(k-1)}} z^{(k)} = diag\left[ \sigma''\left(y^{(k)}\right) \left(\mathbf{W}^{(k)} \partial_{\mathbf{x}_i} z^{(k-1)}\right) \right] \mathbf{W}^{(k)}$.*

**Theorem 2** ($\partial$-CROWN: Linear Bounding $\partial_{\mathbf{x}_i^2} u_\theta$). *Assume that through a previous bounding of $\partial_{\mathbf{x}_i} u_\theta$, we have linear lower and upper bounds on $\partial_{\mathbf{x}_i} z^{(k-1)}$ and $\partial_{z^{(k-1)}} z^{(k)}$. There exist two linear functions $\partial_{\mathbf{x}_i^2} u_\theta^U$ and $\partial_{\mathbf{x}_i^2} u_\theta^L$ such that, $\forall \mathbf{x} \in \mathcal{C}$ it holds that $\partial_{\mathbf{x}_i^2} u_\theta^L \leq \partial_{\mathbf{x}_i^2} u_\theta \leq \partial_{\mathbf{x}_i^2} u_\theta^U$, where the linear coefficients can be computed recursively in closed-form in $\mathcal{O}(L)$ time.*

The formal statement of Theorem 2 and expressions for $\partial_{\mathbf{x}_i^2} u_\theta^L$ and $\partial_{\mathbf{x}_i^2} u_\theta^U$ are in Appendix E.4. As with the first derivative, this bound requires a relaxation of $\sigma''(y^{(k)})$. Note that this also follows a hybrid computation scheme, with the back-propagation and forward substitution paths for bounding $\partial_{\mathbf{x}_i^2} u_\theta$ computations shown in green in Figure 1.

Assuming $\mathcal{C} = \{\mathbf{x} \in \mathbb{R}^{d_0} : \mathbf{x}^L \leq \mathbf{x} \leq \mathbf{x}^U\}$, we can obtain closed-form expressions for constant global bounds on the linear functions $\bar{\partial}_{\mathbf{x}_i} u_\theta^U, \bar{\partial}_{\mathbf{x}_i} u_\theta^L, \bar{\partial}_{\mathbf{x}_i^2} u_\theta^U, \bar{\partial}_{\mathbf{x}_i^2} u_\theta^L$, which we formulate and prove

in Appendix E.5[2]. While here we only compute the expression for the second derivative with respect to the same input, it would be trivial to extend it to cross derivatives (*i.e.*, $\partial_{\mathbf{x}_i \mathbf{x}_j} u_\theta$ for $i \neq j$), as well as to higher order ones.

## 5.2 BOUNDING $f_\theta$

With the partial derivative terms bounded, to bound $f_\theta$, we use McCormick envelopes (McCormick, 1976) to obtain linear lower and upper bound functions $f_\theta^L \leq f_\theta \leq f_\theta^U$: $f_\theta^U = \mu_0^U + \mu_1^U u_\theta + \sum_{j=1}^{r} \sum_{\partial_{\mathbf{x}_i^j} \in \mathcal{N}^{(j)}} \mu_{j,i}^U \partial_{\mathbf{x}_i^j} u_\theta$, and $f_\theta^L = \mu_0^L + \mu_1^L u_\theta + \sum_{j=1}^{r} \sum_{\partial_{\mathbf{x}_i^j} \in \mathcal{N}^{(j)}} \mu_{j,i}^L \partial_{\mathbf{x}_i^j} u_\theta$, where $\mu_0^U$, $\mu_1^U$, and $\mu_{i,j}^U$ are functions of the global lower and upper bounds of $u_\theta$ and $\partial_{\mathbf{x}_i^j} u_\theta$. In the example of Burgers' equation (Equation 2), $f_\theta^U = \mu_0^U + \mu_1^U u_\theta + \mu_{1,0}^U \partial_{\mathbf{x}_0} u_\theta + \mu_{1,1}^U \partial_{\mathbf{x}_1} u_\theta + \mu_{2,1}^U \partial_{\mathbf{x}_1^2} u_\theta$ (and similarly for $f_\theta^L$ with $\mu^L$).

To get $f_\theta^U$ and $f_\theta^L$ as linear functions of $\mathbf{x}$, we replace $u_\theta$ and $\partial_{\mathbf{x}_i^j} u_\theta$ with the lower and upper bound linear expressions from Section 5.1, depending on the sign of the coefficients $\mu^U$ and $\mu^L$. As in Section 5.1, since $\mathcal{C} = \{\mathbf{x} \in \mathbb{R}^{d_0} : \mathbf{x}^L \leq \mathbf{x} \leq \mathbf{x}^U\}$ we can then solve $\max_{\mathbf{x} \in \mathcal{C}} f_\theta^U$ and $\min_{\mathbf{x} \in \mathcal{C}} f_\theta^L$ in closed-form (see Appendix E.5), obtaining constant bounds for $f_\theta$ in $\mathcal{C}$.

## 5.3 TIGHTER BOUNDS VIA GREEDY INPUT BRANCHING

Using $\partial$-CROWN we can compute a bound on a nonlinear of derivatives of $u_\theta$, which we will generally refer to as $h$, for $\mathbf{x} \in \mathcal{C}$. However, given the approximations used throughout the bounding process, it is likely that such bounds will be too loose to be useful when compared to the true lower and upper bound of $h$.

---

**Algorithm 1** Greedy Input Branching

1: **Input:** function $h$, input domain $\mathcal{C}$, # splits $N_b$, # empirical samples $N_s$, # branches per split $N_d$
2: **Result:** lower bound $h_{lb}$, upper bound $h_{ub}$
3: $\mathcal{B} = \emptyset$
4: $\mathcal{B}_\Delta = \emptyset$
5: $\hat{h}_{lb}, \hat{h}_{ub} = \min \setminus \max h(\text{SAMPLE}(\mathcal{C}, N_s))$
6: $h_{lb}, h_{ub} = \partial\text{-CROWN}(h, \mathcal{C})$
7: $\mathcal{B}[\mathcal{C}] = (h_{lb}, h_{ub})$
8: $\mathcal{B}_\Delta[\mathcal{C}] = \max(\hat{h}_{lb} - h_{lb}, h_{ub} - \hat{h}_{ub})$
9: **for** $i \in \{1, \dots, N_b\}$ **do**
10: $\quad \mathcal{C}_i = \mathcal{B}.\text{POP}(\arg\max_{\mathcal{C}'} \mathcal{B}[\mathcal{C}'])$
11: $\quad$ **for each** $\mathcal{C}' \in \text{DOMAINSPLIT}(\mathcal{C}_i, N_d)$ **do**
12: $\quad\quad h'_{lb}, h'_{ub} = \partial\text{-CROWN}(h, \mathcal{C}')$
13: $\quad\quad \mathcal{B}[\mathcal{C}'] = (h'_{lb}, h'_{ub})$
14: $\quad\quad \mathcal{B}_\Delta[\mathcal{C}'] = \max(\hat{h}_{lb} - h'_{lb}, h'_{ub} - \hat{h}_{ub})$
15: $\quad$ **end for**
16: **end for**
17: $h_{lb}, h_{ub} = \min_{\mathcal{C}'} \mathcal{B}_0[\mathcal{C}'], \max_{\mathcal{C}'} \mathcal{B}_1[\mathcal{C}']$
18: **return** $h_{lb}, h_{ub}$

---

To improve these bounds, we introduce *greedy input branching* in Algorithm 1. The idea behind it is to recursively divide the input domain (DOMAINSPLIT, line 9) - exploring the areas where the *current bounds are further from the empirical optima* obtained via sampling (SAMPLE, line 3) - and globally bound the output of $h$ as the worst-case of all the branches (line 13). As the number of splits, $N_b$, increases, so does the tightness of our global bounds. For small dimensional spaces, it suffices to split each branch $\mathcal{C}$ into $N_d = 2^{d_0}$ equal branches. Note that in higher dimensional spaces, a non-equal splitting function, DOMAINSPLIT, can lead to improved convergence to the tighter bounds. The time complexity of greedy input branching is $\mathcal{O}(N_b N_d \mathcal{M})$, where $\mathcal{M}$ is the complexity of running $\partial$-CROWN for each branch. A step-by-step description is provided in Appendix H.

## 6 EXPERIMENTS

The aim of this experimental section is to (i) showcase that the Definition 1 certificates obtained with $\partial$-CROWN are tight compared to empirical errors computed with a large number of samples (Section 6.1), (ii) highlight the relationship of our residual-based certificates and the commonly reported solution errors (Section 6.2, and (iii) qualitatively analyze the importance of greedy input branching in the success of our method (Section 6.3). On top of these results, in Appendix A we study how the training method from Shekarpaz et al. (2022) can lead to a reduction in empirical and certified errors, in Appendix B we empirically show our method is significantly more efficient

---

[2]Note that this is different from the CROWN case in which $\mathcal{C}$ is assumed to be an $\epsilon$-ball around an input $\mathbf{x}$.

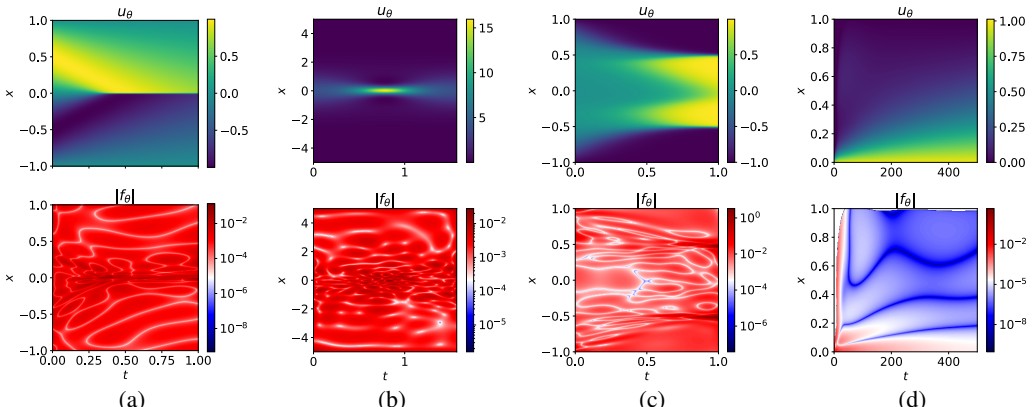

(a)       (b)       (c)       (d)

Figure 2: **Certifying with $\partial$-CROWN**: visualization of the time evolution of $u_\theta$, and the residual errors as a function of the spatial temporal domain (log-scale), $|f_\theta|$, for **(a)** Burgers' equation (Raissi et al., 2019b), **(b)** Schrödinger's equation (Raissi et al., 2019b), **(c)** Allan-Cahn's equation (Monaco and Apiletti, 2023), and **(d)** the Diffusion-Sorption equation (Takamoto et al., 2022).

than Interval Bound Propagation (Gowal et al., 2018; Mirman et al., 2018), while in Appendix C we showcase how $\partial$-CROWN can be used to identify failures in PINN training.

## 6.1 CERTIFYING WITH $\partial$-CROWN

To achieve (i), we apply our post-training certiication framework $\partial$-CROWN to two widely studied PINNs from Raissi et al. (2019a), Burgers' and Schrödinger's equations, as well as to the more complex Allen-Cahn's equation from Monaco and Apiletti (2023), and the Diffusion-Sorption equation from Takamoto et al. (2022). Since $u_\theta$ for these PINNs use $\sigma = \tanh$ activations, we need to be able to linearly relax $\sigma'$ and $\sigma''$ given pre-activation bounds. We propose a practical relaxation in Appendix F. All timing results were obtained on a MacBook Pro with a 10-core M1 Max CPU.

**Burgers' Equation** This one-dimensional PDE is used in several areas of mathematics, fluid dynamics, nonlinear acoustics, gas dynamics and traffic flow, and is derived from the Navier-Stokes equations for the velocity field by dropping the pressure gradient (Raissi et al., 2019a). It is defined on a temporal domain $t \in [0, 1]$ and spatial domain $x \in [-1, 1]$ as:

$$\partial_t u(t, x) + u(t, x)\partial_x u(t, x) - (0.01/\pi)\partial_{x^2} u(t, x) = 0, \tag{3}$$

for $u(0, x) = -\sin(\pi x)$, $u(t, -1) = u(t, 1) = 0$. The solution $u_\theta : \mathbb{R}^2 \to \mathbb{R}$ is modeled by an 8-hidden layer, 20 neurons per layer network (Raissi et al., 2019a). The training process took $\sim 13.35$ minutes, and resulted in a mean $\ell_2$ error of $6.1 \cdot 10^{-4}$, with a visualization in Figure 2a.

**Schrödinger's Equation** Schrödinger's equation is a classical field equation used to study quantum mechanical systems. In Raissi et al. (2019a), Schrödinger's equation is defined with the temporal domain $t \in [0, \pi/2]$ and spatial domain $x \in [-5, 5]$ as:

$$i \, \partial_t u(t, x) + 0.5 \, \partial_{xx} u(t, x) + |u(t, x)|^2 u(t, x) = 0, \tag{4}$$

where $u : [0, \pi/2] \times \mathcal{D} \to \mathbb{C}$ is a complex-valued solution, for initial conditions $u(0, x) = 2 \operatorname{sech}(x)$, and periodic boundary conditions $u(t, -5) = u(t, 5)$ and $\partial_x u(t, -5) = \partial_x u(t, 5)$. As in Raissi et al. (2019b), $u_\theta : \mathbb{R}^2 \to \mathbb{R}^2$ is a 5-hidden layer, 100 neurons per layer network. The training took $\sim 23.67$ minutes, and resulted in a mean $\ell_2$ error of $1.74 \cdot 10^{-3}$, with a visualization in Figure 2b.

**Allan-Cahn Equation** The Allan-Cahn equation is a form of reaction-diffusion equation, describing the phase separation in multi-component alloy systems (Monaco and Apiletti, 2023). In 1D, it is defined on a temporal domain $t \in [0, 1]$ and spatial domain $x \in [-1, 1]$ as:

$$\partial_t u(t, x) + \rho u(t, x)(u^2(t, x) - 1) - \nu \partial_{x^2} u(t, x) = 0, \tag{5}$$

for $\rho = 5$, $\nu = 10^{-4}$, and $u(0, x) = x^2 \cos(\pi x)$, $u(t, -1) = u(t, 1)$. The solution $u_\theta : \mathbb{R}^2 \to \mathbb{R}$ is modeled by an 6-hidden layer, 40 neurons per layer network, and due to its complexity, it is

Table 1: **Certifying with $\partial$-CROWN**: Monte Carlo (MC) sampled maximum values ($10^4$ and $10^6$ samples) and upper bounds computed using $\partial$-CROWN with $N_b$ branchings for ① initial conditions, ② boundary conditions, and ③ residual condition for (a) Burgers (Raissi et al., 2019b), (b) Schrödinger (Raissi et al., 2019b), (c) Allen-Cahn (Monaco and Apiletti, 2023), and (d) Diffusion-Sorption (Takamoto et al., 2022) equations.

| | | MC max ($10^4$) | MC max ($10^6$) | $\partial$-CROWN $u_b$ (time [s]) |
|---|---|---|---|---|
| **(a) Burgers** (Raissi et al., 2019b) | | | | |
| ① | $\|u_\theta(0,x) - u_0(x)\|^2$ | $1.59 \times 10^{-6}$ | $1.59 \times 10^{-6}$ | $2.63 \times 10^{-6}$ (116.5) |
| ② | $\|u_\theta(t,-1)\|^2$ | $8.08 \times 10^{-8}$ | $8.08 \times 10^{-8}$ | $6.63 \times 10^{-7}$ (86.7) |
| | $\|u_\theta(t,1)\|^2$ | $6.54 \times 10^{-8}$ | $6.54 \times 10^{-8}$ | $9.39 \times 10^{-7}$ (89.8) |
| ③ | $\|f_\theta(\mathbf{x})\|^2$ | $1.23 \times 10^{-3}$ | $1.80 \times 10^{-2}$ | $1.03 \times 10^{-1}$ ($2.8 \times 10^5$) |
| **(b) Schrödinger** (Raissi et al., 2019b) | | | | |
| ① | $\|u_\theta(0,x) - u_0(x)\|^2$ | $7.06 \times 10^{-5}$ | $7.06 \times 10^{-5}$ | $8.35 \times 10^{-5}$ (305.2) |
| ② | $\|u_\theta(t,5) - u_\theta(t,-5)\|^2$ | $7.38 \times 10^{-7}$ | $7.38 \times 10^{-7}$ | $5.73 \times 10^{-6}$ (545.4) |
| | $\|\partial_x u_\theta(t,5) - \partial_x u_\theta(t,-5)\|^2$ | $1.14 \times 10^{-5}$ | $1.14 \times 10^{-5}$ | $5.31 \times 10^{-5}$ ($2.4 \times 10^3$) |
| ③ | $\|f_\theta(\mathbf{x})\|^2$ | $7.28 \times 10^{-4}$ | $7.67 \times 10^{-4}$ | $5.55 \times 10^{-3}$ ($1.2 \times 10^6$) |
| **(c) Allen-Cahn** (Monaco and Apiletti, 2023) | | | | |
| ① | $\|u_\theta(0,x) - u_0(x)\|^2$ | $1.60 \times 10^{-3}$ | $1.60 \times 10^{-3}$ | $1.61 \times 10^{-3}$ (52.7) |
| ② | $\|u_\theta(t,-1) - u_\theta(t,1)\|^2$ | $5.66 \times 10^{-6}$ | $5.66 \times 10^{-6}$ | $5.66 \times 10^{-6}$ (95.4) |
| ③ | $\|f_\theta(\mathbf{x})\|^2$ | 10.74 | 10.76 | 10.84 ($6.7 \times 10^5$) |
| **(d) Diffusion-Sorption** (Takamoto et al., 2022) | | | | |
| ① | $\|u_\theta(0,x)\|^2$ | 0.0 | 0.0 | 0.0 (0.2) |
| ② | $\|u_\theta(t,0) - 1\|^2$ | $4.22 \times 10^{-4}$ | $4.39 \times 10^{-4}$ | $1.09 \times 10^{-3}$ (72.5) |
| | $\|u_\theta(t,1) - D\partial_x u_\theta(t,1)\|^2$ | $2.30 \times 10^{-5}$ | $2.34 \times 10^{-5}$ | $2.37 \times 10^{-5}$ (226.4) |
| ③ | $\|f_\theta(\mathbf{x})\|^2$ | $1.10 \times 10^{-3}$ | 21.09 | 21.34 ($2.4 \times 10^6$) |

trained using the Causal training scheme from Monaco and Apiletti (2023). The training process took $\sim 18.56$ minutes, and resulted in a mean $\ell_2$ error of $7.9 \cdot 10^{-3}$, with a visualization in Figure 2c.

**Diffusion-Sorption** The diffusion-sorption equation models a diffusion system which is retarded by a sorption process, with one of the most prominent applications being groundwater contaminant transport (Takamoto et al., 2022). In (Takamoto et al., 2022), the equation is defined on a temporal domain $t \in (0, 500]$ and spatial domain $x \in (0, 1)$ as:

$$\partial_t u(t,x) - D/R(u(t,x))\partial_{x^2} u(t,x) = 0, \tag{6}$$

where $D = 5 \times 10^{-4}$ is the effective diffusion coefficient, and $R(u(t,x))$ is the retardation factor representing the sorption that hinders the diffusion process (Takamoto et al., 2022). In particular, we consider $R(u(t,x)) = 1 + {}^{(1-\phi)}/_{(\phi)}\rho_s k n_f u^{n_f - 1}(t,x)$, where $\phi = 0.29$ is the porosity of the porus medium, $\rho_s = 2880$ is the bulk density, $k = 3.5 \times 10^{-4}$ is the Freundlich's parameter, and $n_f = 0.874$ is the Freundlich's exponent. The initial and boundary conditions are defined as $u(0,x) = 0$, $u(t,0) = 0$ and $u(t,1) = D\partial_x u(t,1)$. The solution $u_\theta : \mathbb{R}^2 \to \mathbb{R}$ is modeled by a 7-hidden layer, 40 neurons per layer network, and we obtain the trained parameters from Takamoto et al. (2022). The mean $\ell_2$ solution error is $9.9 \cdot 10^{-2}$, with a visualization in Figure 2d.

**$\partial$-CROWN certification** We verify the global correctness conditions of the PINNs by applying the framework from Section 5. We report in Table 1 our verification of the initial conditions ① using $N_b = 5k$ splits, boundary conditions ② using $N_b = 5k$ splits, and the certified bounds on the residual condition ③ using $N_b = 2M$ splits. We observe that $\partial$-CROWN approaches the empirical bounds obtained using Monte Carlo sampling while providing the guarantee that no point within the domain breaks those bounds, effectively establishing the tolerances from Definition 1.

6.2 EMPIRICAL RELATION OF $|f_\theta|$ AND $|u_\theta - u|$

One question that might arise from our certification procedure is the relationship between the PINN residual error, $|f_\theta|$, and the solution error with respect to true solution $u$, $|u_\theta - u|$, across the domain. By definition, achieving a low $|f_\theta|$ implies $u_\theta$ is a valid solution for the PDE, but there is no formal guarantee related to $|u_\theta - u|$ within our framework.

Obtaining a bound on $|u_\theta - u|$ is typically a non-trivial task given $u$ might not be unique, and does not necessarily exhibit an analytical solution and can only be computed using a numerical solver. And while some recent works perform this analysis for specific PDEs by exploiting their structure and/or smoothness properties (Mishra and Molinaro, 2022; Ryck and

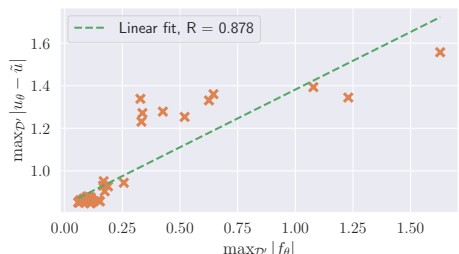

Figure 3: **Residual and solution errors**: connection of the maximum residual error ($\max_{\mathcal{S}'} |f_\theta|$) and the maximum solution error, $\max_{\mathcal{S}'} |u_\theta - \tilde{u}|$, for networks at different epochs of the training process (in orange).

Mishra, 2022; Wang et al., 2022a), these methods typically suffer from scalability and bound tightness issues. As such, we perform an empirical analysis on Burgers' equation using a numerical, finite-difference solver to obtain $\tilde{u}(\mathbf{x})$ for sampled points $\mathbf{x}$. We randomly sample $10^6$ domain points ($\mathcal{S}'$), and compute the maximum residual error, $\max_{\mathbf{x} \in \mathcal{S}'} |f_\theta(\mathbf{x})|$, and the empirical maximum solution error, $\max_{\mathbf{x} \in \mathcal{S}'} |u_\theta(\mathbf{x}) - \tilde{u}(\mathbf{x})|$, for networks obtained at different epochs of the training process. We report the results in Figure 3, with each point corresponding to an instance of a network. As expected, there is a correlation between these errors obtained using a numerical solver, suggesting a similar correlation holds for $|u_\theta - u|$.

## 6.3    ON THE IMPORTANCE OF GREEDY INPUT BRANCHING

A key factor in the success of $\partial$-CROWN in achieving tight bounds of the residual is the greedy input branching procedure from Algorithm 1. To illustrate the fact that a uniform sampling strategy would be significantly more computationally expensive, we plot in Figure 4 the relative density of branches (*i.e.*, the percentage of branches per unit of input domain) in the case of Burgers' and Schrödinger's equations. As can be observed, there are clear imbalances at the level of the branching distribution – with areas away from relative optima of $u_\theta$ being relatively under sampled yet achieving tight bounds – showcasing the efficiency of our strategy.

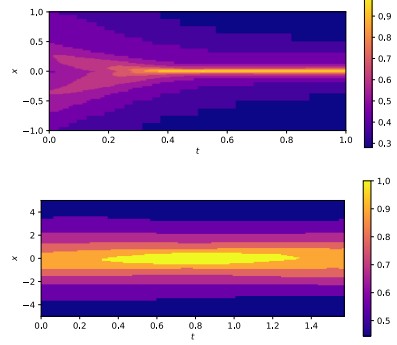

Figure 4: **Branching densities**: relative density of the input branching distribution obtained via Algorithm 1 applied to Burgers' (top) and Schrödinger's (bottom) equations.

## 7    DISCUSSION

We show that $\partial$-CROWN is able to obtain tight upper bounds on the correctness conditions established in Definition 1. Of particular relevance is the case of the residual condition ③ for the Diffusion-Sorption equation, for which varying the number of MC samples leads to distinct results - using $10^4$ estimates puts the maximum at $1.10 \times 10^{-3}$, while $10^6$ samples give an estimate of 21.09 - highlighting the need for our framework to obtain guarantees across the full domain. Note that the absolute values of the residual errors can be seen as a function of the PDE itself, and thus cannot be directly compared across different PINNs. However, in Appendix C we effectively show how $\partial$-CROWN bounds can be used to detect failure cases in PINN training, highlighting another potential use of our framework on top of certifying well-trained ones.

One of the limitations of our method is unquestionably the running time, which for residual verification is in the order of $10^5$–$10^6$ for each of the PINNs studied. This is mainly due to the need to perform a high number of branchings ($2M$) as a result of the looseness of the bounds obtained by $\partial$-CROWN on each individual one. These issues become more accentuated as the input dimension grows, since the number of branches is expected to grow exponentially. In future work we aim to improve the tightness of the bounds to be able to apply our framework to larger, higher dimensional PINNs.

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
