# OpenReview forum: "Efficient Certification of Physics-Informed Neural Networks"
_ICLR.cc/2024/Conference — Submitted to ICLR 2024_

### Official Review · Reviewer_5KvE · 2023-10-27

**Soundness:** 3 good
**Presentation:** 3 good
**Contribution:** 3 good
**Rating:** 5
**Confidence:** 4

**Summary:**

Propose certification for PINN, i.e., guarantees on the worst-case residual error of a PINN, which can be used for prediction of PINN's performance for trustworthy PINN.

**Strengths:**

The paper is well-motivated as the theoretical guarantee is always a huge concern for PINN whose theory is not transparent, making it less practical than traditional finite element methods.

The theory is clearly proved together with supporting experiments.

**Weaknesses:**

The authors only provide theory for 1st and 2nd order derivates in PDEs. However, high-order PDEs are common, such as KdV, Kuramoto-Sivashinsky, and Boussinesq-Burger PDEs.

I don't think the extension to general high-order PDEs is simple, although I noticed the author thought it was trivial, because the expression of high-order derivative is extremely complex, which may affect the tightness of the certification. How does the PDE order affect the tightness of your certification?

Is this framework still valid when PINN encounters propagation failure? See https://arxiv.org/abs/2203.07404 Figure 2. The losses are very small but the error is huge.

To be more specific, there exists a case where PINN converges to a low residual loss but with a huge relative test l2 error, due to the propagation failure, or we can regard it as PINN converges to some spurious local minima corresponding to the trivial solution. In that case, the PINN model will be smooth and low loss. How can your theory deal with this case?

Besides, although the author shows by a figure that the residual loss correlates with the test l2 relative error well, I would like to emphasize that residual loss only provides a loose upper bound for the test l2 relative error. In other words, low residual loss cannot guarantee low test error.

Thus, I suggest the author test their theoretical framework in some tough cases where PINN encounters propagation failure.

**Questions:**

Can this theoretical framework be extended to high-order nonlinear PDEs? High-order PDEs are common, such as KdV, Kuramoto-Sivashinsky, and Boussinesq-Burger PDEs. How does PDE order affect the tightness of your certification?

Is this framework still valid when PINN encounters propagation failure? See https://arxiv.org/abs/2203.07404 Figure 2. The losses are very small but the error is huge.

---

> ### Author Response · Authors · 2023-11-12
> **Authors' Response to Reviewer 5KvE**
>
> We thank the reviewer for the time spent on the paper as well as for their comments.
>
> **(W1) On extending $\partial$-CROWN to certify higher-order PINNs.** We partially agree with the reviewer in that extending $\partial$-CROWN to higher-order derivatives following the more efficient, hybrid scheme that we introduced for the first and second derivatives could be laborious as the derivations of higher-order derivatives become more complicated as a result of the chain rule. However, it is theoretically possible to achieve it by following a similar process.
>
> Following the reasoning from Raissi et al., (2019), we can view higher-order derivatives as a depth-wise extension of the original network (as portrayed also in Figure 1 of our paper). As widely observed by the network verification community, deeper networks are more difficult to verify as bound propagation tends to become looser in incomplete verifiers (Gowal et al., 2018; Wang et al., 2018; Shi et al., 2020; Xu et al., 2020a). We mitigate this problem by introducing Greedy Input Branching to obtain tight bounds on the PINNs used in a more efficient way than other verification methods (see Appendix C). For higher-order PINNs it is likely that one will need (i) tighter relaxations of the nonlinearities of the networks, and (ii) more efficient branching methods that allow us to compensate for the tightness loss in deeper networks.
>
> We will modify the last paragraph of Section 5.1 to clarify these points.
>
> **(W2) On low residual errors and high solution errors.** We thank the authors for the reference to Wang et al. (2022). Regarding the point on Figure 2 the reviewer makes, we would point out that while the residual loss values are of the order of $10^{-2}$, using the causal method the authors of the paper are able to achieve residual loss values below $10^{-5}$ (Figure 4 of the paper), which is what is typically expected in PINN training. This supports the observations we make in Section 6.2. And note also that these values are taken at fixed points instead of taking the maximum across the full domain, which might be significantly higher than at the optimized points.
>
> We partially agree with the reviewer, in that a low maximal residual error does not guarantee a low maximal $\ell_2$ error due to the nonlinearity of the PDEs. We only claim that certified bounds on the boundary, initial conditions and residual error can be interpreted as tolerances in numerical solvers for the PDE – this is explicitly mentioned in Section 6.2.
>
> The reviewer suggests that we “test [our] theoretical framework in some tough cases where PINN encounters propagation failure.” In practice this is what we do in the case of Allen-Cahn’s and the Diffusion-Sorption equations, where the residual errors are so large that we are effectively identifying these PINNs have not been trained properly. We analyze this result in more detail in Appendix C.
>
> We hope these answers are useful to the reviewer in reconsidering the score attributed, and look forward to continuing the discussion.
>
> Additional references:
> - Wang, Sifan, Shyam Sankaran, and Paris Perdikaris. "Respecting causality is all you need for training physics-informed neural networks." arXiv preprint arXiv:2203.07404 (2022).

---

> > ### Author Response · Authors · 2023-11-20
> > **Additional questions or comments**
> >
> > Given the rebuttal process is coming to an end in a couple of days, we hope the reviewer has found our answers to be informative and useful to the reviewing process. If there are any outstanding comments or questions, we are happy to clarify them at this point.

---

> > > ### Author Response · Authors · 2023-11-22
> > >
> > > As the paper scores are currently borderline, we would appreciate the reviewer's input on the rebuttal points now that we are approaching the end of this stage.

---

### Official Review · Reviewer_2eyr · 2023-10-30

**Soundness:** 3 good
**Presentation:** 2 fair
**Contribution:** 3 good
**Rating:** 6
**Confidence:** 3

**Summary:**

The paper provides a definition of global correctness conditions for PINNs that produces the solutions of PINNs. The authors propose CROWN based certification framework to bound first and second derivatives $\mu_\theta$ as well as the $f_\theta$.

While CROWN provides a theoretical bound, given the approximations used throughout the bounding process, such bound is still less tight to bound the actual $h$. The authors utilize a similar idea to BaB, and apply greedy input branching algorithm to the bounding process.

The authors have provided experiments to verify that (1) $\partial$- CROWN are more tight compared to empirical errors computed with a large number of samples (2) residual-based certificates and the commonly reported solution errors are highly correlated. They have also provided an ablation study to check the importance of the branching algorithm.

**Strengths:**

(1) The paper provides a good contribution to the research community. The authors are building the correctness conditions for PINNs over the input domain, which can improve the robustness and trustworthiness of PINN when applying it to the scientific domains.

(2) While CROWN is not new, the authors make novel contributions to apply CROWN to solving PINN's correctness conditions.

(3) In Section 5, the authors have provided detailed theoretical foundations for how to bound the correctness conditions. The proofs in appendix are long but easy to read.  The authors have also provided a theoretical analysis of the running time for $\partial$-CROWM algorithm.

(4) The authors have provided an improved version with greedy input branching, similar to the idea of original branch-and-bound (BaB) process.

**Weaknesses:**

(1) The major concern that reviewer has is that the certification procedure is the on PINN residual error, $f$, but not on the estimation to the true PDE solution. While the authors have provided Exp 2 and Appendix C to demonstrate that there is a strong relationship between the residual error and the PDE solution, it is only conducted on PDEs with a clear PDE solution. More experiments should be conducted with various PDE equations (with multiple solutions, with trivial solutions, etc.) to find out how bounding $f$ is effective. Especially for the recent papers that discuss the failure modes of PINN[1], how would bounding the residual errors help, both theoretically and empirically?

(2) Figure 2 looks meaningless. The Figure 2(a)-2(d) only shows the visualization of the time evolution of $u$, versus the residual
errors for three different PDE equations. However, it is hard to observe any clear patterns of the residual errors, and the authors only briefly mention the residual errors without explaining any scientific insights from the Figure. The reviewer thinks it is better to put the Figure into Appendix, and instead briefly mention Appendix A and B in the main text.

(3) Table 1 shows that  $\partial$-CROWN approaches the empirical bounds obtained using Monte Carlo sampling while providing the guarantee that no point within the domain breaks those bounds. The reviewer think it will be nice to document some time it takes for MC sampling versus the $\partial$-CROWN (at least for residual error), to demonstrate its efficiency.

(4) Figure 1 is occupying too much spaces while the words on the upper right corner are hard to read. Maybe the authors should consider providing a more concise illustration that occupies less space.

(5) In Section 6.3, the authors should consider also including the ablation study on $N_b$ into the main text.

(6) In Section 5.3, the authors mention that "exploring the areas... via sampling (SAMPLE, line 3)", is there a typo and should be line 5 instead of line 3?


[1]: Krishnapriyan, Aditi S., Amir Gholami, Shandian Zhe, Robert M. Kirby, and Michael W. Mahoney. 2021. "Characterizing possible failure modes in physics-informed neural networks." arXiv preprint arXiv:2109.01050.

**Questions:**

(1) In the appendix A, the authors have mentioned to use Physics-informed Adversarial Training to reduce the errors of PINN and demonstrate that $\partial$- CROWN can also bound the residual errors of PIAT. However, there are been various ways of reduce the errors of PINN, for example, scheduled training [1] or building a more complex architecture with transformer [2]. The reviewer is wondering if the $\partial$-CROWN can adapt to NN training that involves more complex loss functions, or more complex architectures?

(2) Similar to Question 1 in Weakness, how could $\partial$-CROWN be used in solving the failures of PINN? If the residual error is large (or being stagnant for a long time), does that imply the PINN is failing to provide a good solution?

(3) Have the authors study the relation between $\partial$-CROWN's time complexity to the size of the PINN's neural network?

(4) How does $\partial$-CROWn behave when the amount of training data is low? Intuitively, the reviewer believes that the amount of training data needs to be evenly spreading across the domain to allow greedy input branching algorithm to work. But what happens when the amount of training data is not evenly spreading?


[1]: Krishnapriyan, Aditi S., Amir Gholami, Shandian Zhe, Robert M. Kirby, and Michael W. Mahoney. 2021. "Characterizing possible failure modes in physics-informed neural networks." arXiv preprint arXiv:2109.01050.

[2]: Zhao, Zhiyuan, Xueying Ding, and B. Aditya Prakash. 2023. "PINNsFormer: A Transformer-Based Framework For Physics-Informed Neural Networks." arXiv preprint arXiv:2307.11833.

---

> ### Author Response · Authors · 2023-11-12
> **Authors' Response to Reviewer 2eyr**
>
> We thank the reviewer for the time spent on the paper as well as for their comments.
>
> **(W1) On the relation of residual and solution error.** This is an important point. We believe the analysis in Section 6.2 and Appendix C indicates a strong correlation between the two types of errors, but there is more to be done in this direction in future work.
>
> **(W2) On Figure 2.** The main purpose of Figure 2 is to give an intuition to readers regarding the landscape of the solution and residual error of each of the PDEs studied. However, we agree with the reviewer that the figure would likely be better suited for the appendix, and will move it there in the final version of the paper.
>
> **(W3) On running time of MC sampling and $\partial$-CROWN.** In general, Monte Carlo sampling is much faster than $\partial$-CROWN for the number of samples we have considered in Table 1 (e.g., inference on $10^6$ samples takes approximately $0.21s$ for Burgers’ equation), scaling linearly with the number of samples for higher values. However, as discussed in the paper, MC sampling only provides a lower bound for the value of the error, whereas $\partial$-CROWN provides a certified upper bound. We will add the runtime information to Table 1.
>
> **(W4) On Figure 1.** We thank the reviewer for the feedback and will consider a few ways of making the figure more compact.
>
> **(W5) Including N_b ablation in the main text.** Given the plan to move Figure 2 to the Appendix as per (W2) of this rebuttal, we will move the ablation on N_b to the main text in the final version of the manuscript.
>
> **(W6) Typo in Algorithm 1.** We thank the reviewer for pointing out that typo which will be corrected in the final version of the paper.

---

> > ### Author Response · Authors · 2023-11-12
> > **Authors' Response to Reviewer 2eyr (2/2)**
> >
> > **(Q1) On verifying different training regimes and architectures.** $\partial$-CROWN is a post-training certification framework, i.e., it’s completely agnostic to the training method. This is why it can be applied to PIAT, and why it can naturally be applied to the curriculum learning scheme proposed by [1], which is similar to the one used for training Takamoto et a., (2022). When it comes to a different architecture such as the transformer one considered in [2], we believe the extension of $\partial$-CROWN to this setting would be doable using a hybrid scheme and by deriving theorems similar to Theorem 1 and 2.
> >
> > **(Q2) Using $\partial$-CROWN to solve PINN failures.** As shown by the examples in Appendix C, $\partial$-CROWN could be used in a post-hoc manner to help identify failure cases of training. A large residual error would suggest that at least in a few points within the continuous domain the solution network does not model the underlying PDE properly. In terms of improving the errors at training time, we believe this is an interesting area of further research.
> >
> > **(Q3) On the relation between $\partial$-CROWN’s time complexity and solution network size.** From a theoretical perspective, the complexity of computing the linear bounds in Theorem 1 and 2 are of the order of $\mathcal{O}(L)$ where $L$ is the number of layers, if we assume a constant number of neurons per layer (otherwise it would be a bilinear dependency on the number of layers and layer dimensionality). From an experimental point of view, we have not run extensive experiments on the scalability on arbitrary networks, but in the table below we show the runtime of bounding the first and second derivatives using $\partial$-CROWN on the full global domain of each PINN as a function of the number of layers and neurons per layer:
> >
> > |  | # layers | # neurons per layer | Total parameters | Runtime [s] |
> > | :--- | :---: | :---: | :---: | :---: |
> > | Burgers’ $\partial_{x} u_\theta$ | 8 | 20 | $1,740$ | $3.7\times 10^{-2}$ |
> > | Allen-Cahn’s $\partial_{x} u_\theta$ | 6 | 40 | $8,320$ | $3.6\times 10^{-2}$ |
> > | Diffusion-Sorption $\partial_{x} u_\theta$ | 7 | 40 | $9,960$ | $5.6\times 10^{-2}$ |
> > | Schrödinger’s $\partial_{x} u_\theta$ | 5 | 100 | $40,800$ | $1.7\times 10^{-1}$ |
> > | | | | |
> > | Burgers’ $\partial_{xx} u_\theta$ | 8 | 20 | $1,740$ | $6.6\times 10^{-2}$ |
> > | Allen-Cahn’s $\partial_{xx} u_\theta$ | 6 | 40 | $8,320$ | $6.2\times 10^{-2}$ |
> > | Diffusion-Sorption $\partial_{x} u_\theta$ | 7 | 40 | $9,960$ | $5.3\times 10^{-2}$ |
> > | Schrödinger’s $\partial_{xx} u_\theta$ | 5 | 100 | $40,800$ | $2.8\times 10^{-1}$ |
> >
> > In practice to achieve the values of Table 1 we used $N_b = 2M$, i.e., we ran bound computations on 2 million subdomains of each PINN’s global domain. Interestingly, we observed that smaller domains induced lower bounding times than the more global ones, mostly due to the nonlinearity relaxations for tanh activations required in Theorem 1 and 2, which we detail in Appendix F. We will include this analysis in the appendix of the final version of the paper.
> >
> > **(Q4) On $\partial$-CROWN on the low training data regime.** While the amount of training data influences the quality of the end solution, we note that $\partial$-CROWN is a post-training framework, so the bounding process occurs successfully regardless of the training procedure. In particular, note that greedy input branching does not make any assumptions on the training data distribution. However, a low or poor distribution of training data often leads to high residual and solution errors – this is observed to a certain extent in Figure 3, where the top right corner points correspond to networks very early in the training process that have not yet been trained properly.
> >
> > We hope these answers are useful to the reviewer, and look forward to continuing the discussion.

---

> > > ### Author Response · Authors · 2023-11-20
> > > **Additional questions or comments**
> > >
> > > Given the rebuttal process is coming to an end in a couple of days, we hope the reviewer has found our answers to be informative and useful to the reviewing process. If there are any outstanding comments or questions, we are happy to clarify them at this point.

---

> > > ### Comment · Reviewer_2eyr · 2023-11-21
> > > **Response to the Authors**
> > >
> > > Thank you for the detailed explanations. However, I am not entirely sure that the authors have answered (W1) On the relation of residual and solution error. It seems that Section 6.2 and Appendix C indicates a strong correlation between the two types of errors, but the paper lacks a theoretical justification to how the two errors are correlated.
> > >
> > > More work should be done towards learning the failure of PINNs and how PINNs could avoid trivial solutions and failure modes in solving the PDEs. The method only provides the post-hoc manner insight, which is less valuable.
> > >
> > > Therefore, I would keep my current scores.

---

### Official Review · Reviewer_uoUZ · 2023-10-31

**Soundness:** 3 good
**Presentation:** 3 good
**Contribution:** 2 fair
**Rating:** 6
**Confidence:** 4

**Summary:**

This paper tries to establish correctness certification for PINN models via the proposed $\partial-$CROWN post-training framework. Under several assumptions, the authors prove two theorems showing that the upper and lower bound of the solution can be computed in $\mathcal O(L)$ time. The authors experimentally demonstrate the tightness of their bound via a few small PDE examples, showing that they can achieve excellent approximation (compared with MC samples) much more efficiently.

**Strengths:**

- Undoubtedly, bounding the error of ML solutions *globally*, as the authors try to tackle, is important and necessary for ML methods to be practically applicable. I am happy to see theoretical and empirical research on this avenue.
- The improved Greedy Input Branching intuitively makes sense to me, although its direction comparison with the non-optimized algorithm seems to be lacking.
- This paper is well-written, and the organization and flow are clear. I also pretty much appreciate the combination of theoretical proof and empirical demonstration.

**Weaknesses:**

While I like this paper, there are several non-negligible problems that have to be addressed before the paper reaches the bar of acceptance.

- First of all, I am not sure why the authors naturally assume that the ground truth can be represented by a FNN. For instance, even a single electron wave function decays exponentially, which an FNN cannot represent. Since the goal here is to bound the global error, I do not think the gap between the best FNN versus the actual ground truth can be ignored. Similarly, assumption 1 limits the applicability of your theorems - what happens if I add some exponential layers, or what if I use other architecture?
- Your theorems only prove the existence of bounds, while your algorithm is still greedy. If I understand correctly, this actually means that: the approximation error (say the numbers in Table 1) is not *guaranteed* to be an upper bound of the global error, right? If so, despite the acceleration you achieve, how can you demonstrate that in a large system, your error approximation is *accurate*?
- I do think the paper should include more complex tasks, as PINN is known to be less accurate in large systems (or long simulation time intervals). Current experiments, while sufficient as a proof of concept, are insufficient to demonstrate the proposed algorithm can indeed provide a better approximation. For instance, you can try multi-body Schrodinger equations, like [1], or HJB equations [2]. These are of large scales and non-linear, making it challenging (IMO) to your method.

[1] Forward Laplacian: A New Computational Framework for Neural Network-based Variational Monte Carlo

[2]  Is $L^2$ Physics-Informed Loss Always Suitable for Training Physics-Informed Neural Network?

**Questions:**

- From my understanding, speed is one of your strengths. Can you more comprehensively compare with previous methods to demonstrate how fast your method is? A table or figure will be ideal.
- I think the paper could be benefited by including more takeaways, especially for the theorems. I can sort of imagine how this theorem is proved, but it would be great if the authors could give a proof sketch and discuss how each assumption and condition works.
- notation layer $k$: try to use $l$ since the total layer number is $L$.

---

> ### Author Response · Authors · 2023-11-12
> **Authors' Response to Reviewer uoUZ**
>
> We thank the reviewer for the time spent on the paper as well as for their comments.
>
> **(W1) On the assumptions and applicability of $\partial$-CROWN.** Regarding the reviewer’s first point on the representation capacities of feedforward networks, we note that these networks are, in theory, universal approximators in the infinite width limit, so they should be able to represent arbitrary solution functions given an appropriate choice of activation. However, we agree with the reviewer in that it might not be an optimal architecture choice for some PINNs (which is likely the reason behind at least some training failures). Despite this, it is widely used in the previous literature, and hence why we chose to focus on these types of networks.
>
> Note that some previous works providing guarantees for PINNs such as Shin et al., (2020), Wang et al., (2022b), or Mishra and Molinaro (2022) only provide them for specific families of PINNs, whereas our work is more general since it can be applied to all PINNs represented by FNNs.
>
> **(W2) On the greedy nature of our method.** Our method provides a ***guaranteed upper bound*** on the global errors over a domain – so the upper bounds provided in Table 1 are guaranteed bounds. We introduce greedy input branching in Section 5.3. to improve the tightness of the bounds obtained using $\partial$-CROWN in the full PDE domain by splitting it into subdomains. Since the union of all the subdomains corresponds to the full PDE domain, taking the worst-case bound of each of the subdomains gives us a bound on the error in the full domain. As such, the bounds obtained via greedy input branching are **guaranteed** to be globally satisfied over the domain - there is no "approximation" that breaks the bound. More details on the procedure of greedy input branching is available in Appendix H.
>
> **(W3) On further experiments.** The PINNs in the experiments section were chosen primarily because they are well established in previous literature in the field, from the classic Burgers’ and Schrödinger’s equation in Raissi et al. (2019a) to more recent and difficult ones to learn in the Allen-Cahn’s equation from Monaco and Apiletti (2023) and the Diffusion-Sorption from Takamoto et al. (2022). The HJB equation from [2] is modeled as an MLP, so our framework could be applied to this problem, but as the reviewer mentions the scale of the network would be challenging for $\partial$-CROWN as it is over $1,500\times$ bigger than the networks considered. This drawback of our method is explicitly mentioned in the Section 7 of the paper, and we hope future work will help scale verification methods to those larger networks.
>
> **(Q1) On the efficiency of our method.** To the best of our knowledge, $\partial$-CROWN is the first framework designed to bound the errors of general PINNs. However, for completeness of analysis, in Appendix B we extend Interval Bound Propagation (IBP) (Gowal et al., 2018; Mirman et al., 2018) – known for its simplicity and trading-off bound tightness for speed – to this setting. The results are presented in Table 3 of our paper (Appendix B). It presents the performance of IBP and $\partial$-CROWN on the initial, boundary and residual errors for a fixed runtime limit in Burgers’ equation. This is a fair comparison which takes into account the runtime/tightness trade-off of the two methods. We observe that $\partial$-CROWN is significantly more efficient than IBP, achieving bounds that are $165 - 1,566\times$ tighter than the baseline in the same total runtime.
>
> **(Q2) On takeaways for Theorem 1 and 2.** While the expressions for the linear lower and upper bounds are quite efficient to compute in practice, we found it difficult to include them in the main text without excessive notation. However, we agree that the dependencies of the linear functions are not explicit in the theorem, and that a proof sketch should be added in the following paragraph. We will make these modifications in the final version of the paper.
>
> **(Q3) On notation.** We thank the reviewer for pointing this out, and we will make the suggested adjustment in the final version of the paper.
>
> We hope these answers are useful to the reviewer in reconsidering the score attributed, and look forward to continuing the discussion.

---

> > ### Author Response · Authors · 2023-11-20
> > **Additional questions or comments**
> >
> > Given the rebuttal process is coming to an end in a couple of days, we hope the reviewer has found our answers to be informative and useful to the reviewing process. If there are any outstanding comments or questions, we are happy to clarify them at this point.

---

> > ### Comment · Reviewer_uoUZ · 2023-11-20
> >
> > Thanks for your reply.
> >
> > - **(W1) On the assumptions and applicability of CROWN.** I don't think the universality expression explanation is convincing enough for not considering a wider range of network architectures. In fact, despite the universality, it's CNN (not FNN) that works well in images, and it's Transformer (not FNN) that works well in languages. I understand that most previous works consider even limited structures, but what's the challenge that prevents your current methods from being applied to other architectures? Is it because some of the mathematical techniques you use work only on linear (or piecewise linear) models? That's the reason why I want to learn more about the proof sketch. More explanation about why there is a limitation on the application range is important, and I encourage the authors to add this.
> >
> > - **(W2) On the greedy nature of our method.** Thanks. That makes sense.
> >
> > I am happy to increase my rating if the authors can address the (W1) issue, either by extending to a more general architecture, or by explicitly pointing out what limits your method.

---

> > > ### Author Response · Authors · 2023-11-21
> > > **Response to Comment by Reviewer uoUZ**
> > >
> > > We thank the reviewer for engaging in the rebuttal.
> > >
> > > Regarding (W1), we agree with the reviewer that the universality argument is not enough to justify not using other architectures, and we will attempt to clarify why our bounds in Theorem 1 and 2 are only applicable to FNNs.
> > >
> > > Our work is based on the established field of convex relaxations of the nonlinearities of the networks, and bound propagation first introduced in Zhang et al., (2018) which applied these techniques to FNNs. The main idea is simple: for the purpose of bounding the output of each network layer in an FNN, instead of modeling the nonlinear output of the activation they instead lower and upper bound it using linear functions. Assuming we have relaxed each layer, the output of the network can be bounded by linear functions of the input (for a bounded input) by *back-propagating* the outputs of each layer w.r.t. their inputs until we reach the input layer. For more details on this mechanism, we refer to Zhang et al., (2018). As shown in Xu et al., (2020a), this can be generalized to arbitrary computation graphs by relaxing the output of every nonlinearity in its path instead of just the nonlinear activations of the FNN. As mentioned in our work, the issue with applying this approach from Xu et al., (2020a) directly is its computational complexity.
> > >
> > > $\partial$-CROWN's bounds on the first and second derivative in Theorem 1 and 2 make use of the explicit structure of the derivative networks computed in Lemma 1 and 2, respectively. **The fact that these are FNNs is explicit in Lemma 1 and 2, and is used throughout the proofs of the bounds of Theorem 1 and 2. As a result, these bounds are only applicable to these types of networks**. This is done to permit a *hybrid* bound propagation in Theorem 1 and 2 as depicted in Figure 1 where at points the back-propagation of bounds present in Zhang et al., (2018) and Xu et al., (2020a) is replaced by a forward substitution in the bounds - see also Equation 10 in Appendix E.3. for the forward substitution in the proof of Theorem 1; and the bounds on $\sigma''(y_{j}^{(k)})$ and Equation 23 in Appendix E.4. in the proof of Theorem 2. This achieves a lower computational complexity compared to the method from Xu et al., (2020a) and allows the method to scale to the PINNs in this paper.
> > >
> > > As such, the reviewer is correct in saying the derived method can only be applied to FNNs due to the mathematical techniques used for bounding. In practice we did not concern ourselves with other architectures as FNNs are the most widely used in PINNs - yet extending this to other networks (e.g., including exponential layers) should be matter of modifying Lemma 1 and 2, and to generate bounds in a similar fashion to Theorem 1 and 2. We will modify the main text of the paper to explicitly mention these intuitions in a proof sketch of Theorem 1 and 2.
> > >
> > > We hope this response has clarified (W1) for the reviewer.

---

> > > > ### Comment · Reviewer_uoUZ · 2023-11-22
> > > >
> > > > Thanks for adding this to the paper. I will increase my rating to 6.

---

### Official Review · Reviewer_3Mdo · 2023-10-31

**Soundness:** 3 good
**Presentation:** 2 fair
**Contribution:** 2 fair
**Rating:** 5
**Confidence:** 3

**Summary:**

This paper proposes to solve the question of “efficient certification” of PINNs which – as the authors state – is the question of trying to quickly estimate bounds on the supremum (over domain points) error made by the neural surrogate for the residuals. The authors give experiments to suggest that their methods give upperbounds almost as good as the estimations made via random sampling of domain points.

**Strengths:**

The claims that have been proven in this paper do indeed seem a bit surprising – that upto second order of derivatives for the neural net, these functions can be linearly bounded bothways.

**Weaknesses:**

Prima facie the paper is very badly written – to the point of being very much devoid of motivations. Till one gets to the experiments, it's entirely unclear what is the “certification” question that is even being solved. In fact even after reading everything in the main paper, it's not clear at all as to why the random point sampling way is not enough to implement a check of the inequalities as required in Definition 1. What exactly is being gained by this method (as demonstrated in the third column of Table 1) as opposed to the results in the first two columns?

Even the run-time reported to obtain some of these numbers in the third column of Table 1 seems to be in millions of seconds.
That is more than a day! So this method isn't hinged on an argument of speed either.

Secondly, I am not able to ascertain from the given descriptions if the presence of these linear two-way bounds is a special case for the first two derivatives of a neural net or does this happen for all derivatives. Is the method $\partial-$CROWN hinged on the validity of these linear bounds? In that case, why is the method of interest to the general PINN formalism where arbitrarily high orders of derivatives can potentially occur.

Thirdly, the theorems are very badly stated. It's not clear from their statements as to whether or not there is an explicit expression for these linear bounds and if yes (which it seems to be from the appendix!) then what information is required to be able to compute them.

Fourthly, the choice of the PDE examples for testing are very skewed. If one is doing experiments on multiple PDEs then it would have been better to include PDEs that have more than 2 variables. It does not help to have all the examples be stuck to two variable PDEs!

Also, the Schrodinger PDE example looks a bit strange. Which natural Hamiltonian function even leads to this? This seems to be coming from a system with a potential energy which is “$-|u|^2$” - and that is never a natural setting – to the best of my knowledge of quantum theory!

**Questions:**

It would be great if the authors can give a precise answer to the first two weaknesses pointed out above.

---

> ### Author Response · Authors · 2023-11-12
> **Authors' Response to Reviewer 3Mdo**
>
> We thank the reviewer for the time spent on the paper as well as for their comments.
>
> **(W1) On the motivation of PINN certification and efficiency.** We stress in the introduction that the main goal of this paper is to provide a formal guarantee for *every* input within the feasible domain of the PINN. While most previous works report computed mean and/or max $\ell_2$ solution error computed over a finite number of points, this cannot be relied on in practice as they might not sample domain points where the error is large. We showcase this in the Diffusion-Sorption example in Table 1, where the residual bound estimate using $10^4$ MC samples is $1.1\times 10^{-3}$, while the certified upper bound is obtained at $21.34$. **Through sampling we are only able to obtain lower bound estimates on the error** which, as we can see from Table 1, can be quite distant from the actual error at specific domain points (visible on the same equation when we increase the number of samples from $10^4$ to $10^6$). This highlights the need for **certified upper bounds, which we provide with $\partial$-CROWN**.
>
> As far as we are aware, our framework is the first one that can be used to certify arbitrary PINNs with solutions modeled by fully connected neural networks. And while runtime is still a drawback of our framework, in Appendix B we perform an efficiency comparison with an adaptation of Interval Bound Propagation (IBP) (Gowal et al., 2018; Mirman et al., 2018) – known for its simplicity and trading-off bound tightness for speed – to this setting. We compare the performance of IBP and $\partial$-CROWN on the initial, boundary and residual errors for a fixed runtime limit in Burgers’ equation. This is a fair comparison which takes into account the runtime/tightness trade-off of the two methods. We observe that $\partial$-CROWN is significantly more efficient than IBP, achieving bounds that are $165 - 1,566\times$ tighter than the baseline in the same total runtime.
>
> **(W2) On the linear bounds and higher-order derivatives.** As we note at the beginning of page 6 of the manuscript, $\partial$-CROWN can be extended to higher-order derivatives in a similar fashion to the process used for deriving the second-order bounds. As such, it is ***not a special case of second-order PINNs***, and it can be applied to higher-order ones too. The linear bounds are obtained by relaxing the nonlinearities using linear overapproximations, so they can be applied to any function defined over a continuous domain.
>
> **(W3) On theorem notation.** The bounds obtained in Theorem 1 and 2 can be computed from the weights of the solution network $u_\theta$ (that is $W^{(k)}$) and the bounds provided by Assumption 1 ($\mathbf{A}^{(k),L}$, $\mathbf{a}^{(k),L}$, $\mathbf{A}^{(k),U}$, $\mathbf{a}^{(k),U}$, $y^{(k), L}$ and $y^{(k), U}$), using the recursive definition provided in Appendix E. While these expressions are quite efficient to compute in practice, we found it difficult to include them in the main text without excessive notation. However, we will change Theorem 1 and Theorem 2 to clarify the dependencies of the linear bounds.
>
> **(W4) On the choice of PINNs.** The PINNs in the experiments section were chosen primarily because they are well established in previous literature in the field, from the classic Burgers’ and Schrödinger’s equation in Raissi et al. (2019a) to the more recent and difficult ones to learn in the Allen-Cahn’s equation from Monaco and Apiletti (2023) and the Diffusion-Sorption from Takamoto et al. (2022). Another important factor in the choice was the availability of either the code or trained models from all of these previous works. While we considered other suitable higher dimensional PINNs, such as several of the Navier-Stokes equations from Jin et al., (2021), or the Gray-Scott system from Giampaolo et al., (2022), neither training code nor the pre-trained models were released that allow us to apply $\partial$-CROWN.
>
> We hope these answers are clear and useful to the reviewer in reconsidering the score attributed, and look forward to continuing the discussion.
>
> Additional references:
> - Giampaolo, Fabio, et al. "Physics-informed neural networks approach for 1D and 2D Gray-Scott systems." Advanced Modeling and Simulation in Engineering Sciences 9.1 (2022): 1-17.
> - Jin, Xiaowei, et al. "NSFnets (Navier-Stokes flow nets): Physics-informed neural networks for the incompressible Navier-Stokes equations." Journal of Computational Physics 426 (2021): 109951.

---

> > ### Author Response · Authors · 2023-11-20
> > **Additional comments or questions**
> >
> > Given the rebuttal process is coming to an end in a couple of days, we hope the reviewer has found our answers to be informative and useful to the reviewing process. If there are any outstanding comments or questions, we are happy to clarify them at this point.

---

> > > ### Author Response · Authors · 2023-11-22
> > >
> > > As the paper scores are currently borderline, we would appreciate the reviewer's input on the rebuttal points now that we are approaching the end of this stage.

---

### Meta-Review · Area_Chair_KxBX · 2023-12-11

**Metareview:**

The paper considers the problem of providing **guaranteed** upper bounds on the quality of a solution to a PINN problem: that is, a certificate for the upper bound of the loss, and the amount of violation of the PDE constraints. The way the paper proceeds is to take both these quantities, and track 0-th order quantities using prior work (CROWN), and track 1st and 2nd order quantities by applying the chain rule without a quadratic runtime blowup (similarly to how it's done in backprop/autodifferentiation). They also provide some heuristics for a branch-and-bound approach to speed up the method.

The reviews are borderline, and my opinion is somewhat similar. In terms of technical novelty, the method is somewhat incremental over CROWN. In terms of runtime, the method is still quite expensive, even on relatively simple PDEs. The evaluation is rather limited --- even the PDEs termed as "more challenging" (Allan-Cahn and Diffusion-Sorption) are still relatively "nice". For instance, it'd be nice to test some version of Navier-Stokes where the solutions are much less stable in time. It would also be nice to have a more extensive study on how well certifying the *constraints/loss* translates to *being close to the solution*. There is some discussion on this in the paper, but it's relatively limited.

**Justification For Why Not Higher Score:**

The paper is relatively incremental on a technical level; the evaluation is relatively limited. There is relatively little work / discussion on what certifying the constraints/loss implies about proximity to the solution.

**Justification For Why Not Lower Score:**

N/A

---

### Decision · Program_Chairs · 2024-01-16

Reject